# Liposomal Irinotecan for Treatment of Colorectal Cancer in a Preclinical Model

**DOI:** 10.3390/cancers11030281

**Published:** 2019-02-27

**Authors:** Jiao-Ren Huang, Mei-Hsien Lee, Wen-Shan Li, Han-Chun Wu

**Affiliations:** 1Ph.D. program for Cancer Molecular Biology and Drug Discovery, College of Medical Science and Technology, Taipei Medical University and Academia Sinica, Taipei 110, Taiwan; jaran0707@gmail.com; 2Institute of Cellular and Organismic Biology, Academia Sinica, Taipei 115, Taiwan; 3Graduate Institute of Pharmacognosy, College of Pharmacy, Taipei Medical University, Taipei 110, Taiwan; lmh@tmu.edu.tw; 4Institute of Chemistry, Academia Sinica, Taipei 115, Taiwan; wenshan@gate.sinica.edu.tw

**Keywords:** colorectal cancer, liposomal irinotecan, combination targeted therapy, drug delivery system, microbiota

## Abstract

Colorectal cancer (CRC) is the most frequently diagnosed cancer and leading cause of cancer-related deaths worldwide. Because of the use of first-line CRC treatments, such as irinotecan (IRI), is hindered by dose-limiting side effects, improved drug delivery systems may have major clinical benefits for CRC treatment. In this study, we generate and characterize liposomal irinotecan (Lipo-IRI), a lipid-based nanoparticle, which shows excellent bioavailability and pharmacokinetics. Additionally, this formulation allows IRI to be maintained in active form and prolongs its half-life in circulation compared to IRI in solution. Compared with IRI statistically, the level of prostaglandin E2 (PGE2) in colonic tissue decreases, and *Bifidobacterium* spp. (beneficial intestinal microbiota) content increases in the Lipo-IRI-treated group. Moreover, no damage is observed by the hematoxylin and eosin staining of the normal tissue samples from the Lipo-IRI-treated group. In a xenograft mouse model, CRC tumors shrink markedly following Lipo-IRI treatment, and mice receiving a targeted combination of Lipo-IRI and liposomal doxorubicin (Lipo-Dox) extend their survival rate significantly. Overall, our results demonstrate that this formulation of Lipo-IRI shows a great potential for the treatment of colorectal cancer.

## 1. Introduction

Colorectal cancer (CRC), also known as bowel cancer, is the third most common malignant disease worldwide [1]. The disease usually develops slowly, and most cases are adenocarcinomas [2]. Screening of at-risk individuals can help to detect CRC at early stages, which greatly improves the prognosis and decreases the CRC mortality. Indeed, if cancerous polyps are detected early, the five-year survival rate after treatment is over 90% [3]. Unfortunately, the majority of colon cancer patients are diagnosed with a late-stage disease, when the survival rates are much lower. Generally, patients suffering from primary CRC will undergo a surgical resection of local tumors. However, physicians will sometimes suggest a combination treatment regimen, which includes either chemotherapy or radiotherapy. In such cases, the first-line chemotherapeutic drug for CRC patients is irinotecan (IRI or CPT-11), an analog of camptothecin (CPT).

CPT is a quinoline-based alkaloid derived from the bark of the Camptotheca acuminate tree [4], which is native to Asia. The first clinical trials with CPT in the 1970s showed that the drug had a promising anti-tumor activity. However, its effectiveness was limited by its poor aqueous solubility and severe side effects [5,6]. IRI is one of these analogues, which acts as a pro-drug of SN-38, a carboxylesterase-generated active metabolite. IRI/SN-38 stabilize Topo I-DNA adducts by binding covalently to nicked DNA, resulting in irreversible double-strand breaks [7,8]. IRI has five planar heterocyclic structures with α-hydroxylactone in its E-ring, which interacts with carboxylesterase and is essential for anti-tumor activity. The anti-tumor activity of IRI is enhanced at a low pH (Appendix A), as IRI undergoes a pH-dependent reversible hydrolysis from its potent lactone form (pH < 7, active) to a much less potent carboxylate form (pH > 7, inactive). Therefore, the therapeutic action of IRI is dependent on the integrity of the E-ring structure [9], and effective drug formulations will maintain this moiety.

In addition to CRC, IRI has also shown promising effects in lung, breast and ovarian cancers [10,11]; however, the drug exhibits side effects in any application. For example, IRI-associated diarrhea may be severe and clinically significant, sometimes leading to life-threatening dehydration that requires hospitalization [12,13,14]. Moreover, IRI may cause immune-suppression, leading to thrombocytopenia, neutropenia or anemia [15,16], and other common side effects that include hair loss, mouth sores or ulcers. Although IRI can efficiently inhibit the growth of cancer cells, its toxic effects on healthy organs and normal cells still limit its therapeutic utility.

Improvements in cancer therapies mostly result from the development of new therapeutic molecules, combinations of treatments or the development of novel formulations. Since cancer patients who receive chemotherapies often experience poor outcomes that are related to severe side effects, a large number of polymer- or liposome-based carriers have been developed in an attempt to improve the pharmacokinetics and bio-distribution of active therapeutics. Currently, there are numerous liposomal drugs that are either approved for clinical use or are in clinical trials with promising results [17,18,19,20,21,22]. Up to now, hundreds of drugs have been incorporated into liposomes of various sizes, compositions and surface characteristics. Liposomes are able to encapsulate hydrophilic drugs inside the aqueous inner core or hydrophobic drugs within the lipid bilayer. Moreover, a site-specific delivery to solid tumors by targeting liposomes can facilitate a slow drug release at the site of the tumor, thus increasing the drug concentration in the solid tumor relative to other tissues [23,24,25,26]. Notably, these modern drug delivery systems may be utilized for more than one anti-tumor drug to improve efficacy and provide valuable information for further preclinical and clinical studies.

In this study, we encapsulated high concentrations of IRI in liposomes (Lipo-IRI), using a thin-film hydration method, and the stability was evaluated by examining the z-average, zeta potential and polydispersity index (PDI). By packaging drugs in liposome particles, the side effects were reduced and the bioavailability of the anti-cancer drugs was improved in our novel Lipo-IRI formulation. We also assessed the cell cytotoxicity in vitro, while the pharmacokinetics and bio-distribution were assessed in vivo. In order to evaluate the drug safety in mice treated with free IRI or Lipo-IRI, we compared several parameters that are closely related to clinically relevant side effects. Finally, the therapeutic efficacy of Lipo-IRI was evaluated in tumor-bearing mice, and a combination targeted therapy through colorectal cancer-targeted peptide, pHCT74 [26], was tested by the co-administration of pHCT74-Lipo-IRI and pHCT74-Lipo-Dox in a large-tumor xenograft model. Our results revealed that this new formulation of Lipo-IRI has markedly increased the pharmacokinetics and drug delivery into the tumor tissue, thereby significantly increasing the therapeutic index compared with free IRI.

## 2. Results

### 2.1. Characterization of Liposomal Irinotecan (Lipo-IRI)

The z-average and size distribution of the Lipo-IRI solution were determined by Dynamic Light Scattering. The hydrodynamic diameter of Lipo-IRI ranged from 100 to 110 nm (Figure 1A), and the particle size, as determined by Nanosight NS3000, was similar at 105 nm (Figure 1B). Additionally, the polydispersity index (PDI) was less than 0.1, indicating that Lipo-IRI was mono-dispersed and homogeneous. The Lipo-IRI particle concentration was estimated with a Nanosight NS3000 and quantified by NTA 3.0 software. Approximately 64,000 ± 8900 IRI molecules were entrapped within a single liposome in a precipitated form. In addition, we evaluated the size, entrapped drug and morphology of the liposomal drug by cryogenic transmission electron microscopy (cryo-TEM). The encapsulation efficiency was estimated to be above 97%. The cryo-TEM images of Lipo-IRI and the empty liposome were compared (Figure 1C,D). The liposome particle sizes were around 105 nm, and the IRI precipitates were seen as dark areas inside the homogeneous liposome solution.

### 2.2. Stability of Lipo-IRI

The zeta potential and z-average were measured over time by DLS (Figure 2A,B). The z-average and PDI were found to be stable even after storage for 400 days at 4 °C. Since no significant degradation was observed within 400 days, we concluded that the formulation of Lipo-IRI was quite stable.

Because equilibrium favors the formation of inactive/less toxic IRI carboxylate (IRI^C^) at a physiological pH or higher, the IRI encapsulation in acidic conditions within the liposome enhanced the amount of active lactone (IRI^L^) and extended the IRI^L^ to IRI^C^ transformation time (Figure 2C). In order to investigate the effect of the environmental pH on the Lipo-IRI drug release, a dialysis assay was performed. After Lipo-IRI was dialyzed against a HEPES buffer at 37 °C for 72 h, more than 90% of IRI was still retained in the liposome, and less than 10% of the drug was released (Figure 2D). On the contrary, free IRI was quickly released into the dialyzing medium within 24 h. Therefore, Lipo-IRI may delay and extend the release time after administration. Dialysis was also performed in a more acidic buffer to mimic the drug release in endosomes or lysosomes. As shown in Figure 2D, IRI was released from liposomes and detected after only 1 h of dialysis in a pH 4.0 buffer.

### 2.3. In Vitro Cell Viability of Irinotecan

IRI is a prodrug, which can be converted to 7-ethyl-10-hydroxycamptothecin (SN-38) by carboxylesterase enzyme [27,28]. Although SN-38 possesses more potent anti-tumor activity than IRI, it also shows an increased toxicity and harmful side effects. We screened several tumor cell lines and found that IRI (or SN-38) was more toxic to HCT 116, SK-HEP-1 and A549 cell lines, which are colon, liver and lung cancer cell lines, respectively. To determine the sensitivity of tumor cell lines to IRI, the 3-(4,5-dimethylthiazol-2-yl)-2,5-di-phenyl-tetra-zoluim bromide (MTT) assay was performed and dose-response curves were generated to determine IC_50_ values. As shown in Appendix A, the IC_50_ of IRI was approximately 25.37 μM, 13.58 μM and 36.29 μM in the HCT 116, SK-HEP-1 and A549 cell lines, respectively. However, the copper-based liposomes did not cause any inhibition of cell growth.

### 2.4. Chromatographic Analysis of Lipo-IRI

The IRI^L^ conformation was found to be unstable due to hydrolysis. Therefore, in order to prevent degradation, the samples were rapidly frozen. The total concentration of IRI (IRI^L^ and IRI^C^) was measured and the lactone formation ratio (IRI^L^/total IRI) was more than 0.97. A fluorescence detector with an excitation wavelength of 375 nm and emission wavelength of 500 nm yielded a good signal-to-noise ratio for the compounds. The total run time was 15 min. The retention times of the carboxylate and the lactone forms were 4.30 min and 9.30 min, respectively. The lactone form of SN-38, which is a metabolite of IRI^L^, was eluted at 10.45 min (Appendix A). A calibration curve for IRI, either in lactone form or total form, was obtained by plotting the peak areas from the fluorescence detector. A linear regression of the calibration curve yielded the equation Y = 9.610 × 10^6^X + 7.631 × 10^6^ (R^2^ = 0.999) (Appendix A).

### 2.5. Plasma Pharmacokinetics and Bio-Distribution of Lipo-IRI

The time profiles of the plasma drug concentrations for free IRI or Lipo-IRI were determined. Circulating IRI was undetectable within 1 h after the administration of free IRI, suggesting that free IRI was rapidly eliminated from the blood stream. On the contrary, the concentration of Lipo-IRI achieved 881.71 μg/mL at 10 min post-administration, which was 130-times higher than that of free IRI (6.87 μg/mL). After an analysis using GraphPad Prism 6.0 software, the half-life T_1/2_ of Lipo-IRI was determined to be 5.33 hours, which was 70-times longer than that of free IRI. The area under the curve (AUC) for Lipo-IRI was 6155 μg·h/mL. The main pharmacokinetic parameters are shown in Figure 3A. The formulation of IRI in the liposomes not only extended the half-life, but it also increased the AUC of the drug.

A bio-distribution study was performed by analyzing organs from tumor-bearing mice that had received free IRI or Lipo-IRI. Following a PBS perfusion, several organs, including the brain, heart, lung, liver, kidney, spleen, colon and tumor, were collected at various times post-injection. The tissue was homogenized and quantified by a reverse phase high performance liquid chromatography (RP-HPLC). The mononuclear phagocyte clearance organs, such as the liver, kidney and spleen, had a higher accumulation of drug (Figure 3B,C). The AUC for the tumors in the Lipo-IRI group (82.04 μg·h/mL) was 4-fold higher than that in the IRI group (20.69 μg·h/mL) (Figure 3D).

### 2.6. In Vivo Therapeutic Efficacy of Liposomal Irinotecan

IRI is mainly activated to SN-38 by the carboxylesterase enzyme. According to the MTT assay, three tumor models (colon, liver and lung) were demonstrated to be more sensitive to IRI. Lipo-IRI produced a tumor response in vivo in all three of these three xenograft models (Appendix A). After the administration of Lipo-IRI at 10 mg/kg twice per week for a total of eight injections (twice weekly × 4), the tumor volume shrank dramatically compared to the PBS-treated group, and the tumor growth inhibition rates were calculated to be 98.35%, 95.77% and 87.40% in the HCT 116, SK-HEP-1 and A549 xenograft models, respectively.

Additionally, we evaluated the therapeutic efficacy of Lipo-IRI in HCT 116 tumor xenograft models. When the tumor sizes reached 100 mm^3^, the mice were injected with either free IRI or Lipo-IRI at 2 or 5 mg/kg (twice weekly × 4). There was no significant difference in tumor growth between the free IRI- and PBS-treated groups. Interestingly, the mice receiving Lipo-IRI at 2 mg/kg exhibited a marked tumor growth delay, resulting in 67.14% inhibition compared to the PBS-treated controls (Figure 4A). Body weight changes were monitored during the drug treatment and the tumor mass was measured after sacrificing the mice (Figure 4B,C). Surprisingly, the tumor growth inhibition (TGI) reached 95.2% or 97.3% with significant reductions in the tumor mass when the tumor-bearing mice were treated with 5 or 10 mg/kg Lipo-IRI (Figure 4D). The tumor mass decreased rapidly in the 5 mg/kg Lipo-IRI-treated group, and there was no obvious difference between the 5 and 10 mg/kg Lipo-IRI-treated groups (Figure 4F,G). In addition, there was no significant change in the body weight of any group (Figure 4E). Similar results were observed in the SW620 xenograft model: the tumor bearing mice received IRI or Lipo-IRI twice weekly, and the mice that were administered Lipo-IRI at 5 mg/kg had a marked tumor reduction (TGI was 88.6%) (Figure 4H).

### 2.7. Side Effects of Irinotecan

The tumor bearing mice were treated with drugs at a dose of 5 mg/kg for eight injections and euthanized on day 28. The whole blood and serum were collected for complete blood count and toxicity analyses. The administration of Lipo-IRI or free IRI showed no obvious effects on the levels of white blood cells (WBCs), red blood cells (RBCs) or neutrophils (Figure 5A–C). However, the monocyte and lymphocyte levels were less affected in mice that were treated with Lipo-IRI in comparison to free IRI (Figure 5D,E) but no difference in neutrophil level (Figure 5F). No significant hepatotoxicity (aspartate aminotransferase (AST) and alanine aminotransferase (ALT)) or nephrotoxicity (creatinine and blood urea nitrogen) were observed at 5 mg/kg treatments (Figure 5G–J). No histological abnormalities were found in the vital organs, including the brain, heart, liver, kidney and spleen. There were mild erosions and a slight lymphocytic infiltration in the lamina propria of the colon after the treatment with either IRI or Lipo-IRI, but no damage to the crypts was found in the histological analysis (Figure 5K).

Mice were treated with the drug twice weekly over four weeks. The colons were harvested, and the levels of PGE2, myeloperoxidase (MPO) and tumor necrosis factor α (TNF-α) were estimated by ELISA. The PGE2 levels were attenuated in the Lipo-IRI-treated group compared to the IRI-treated group (Figure 6A). There was no significant difference in the levels of MPO or TNF-α between the Lipo-IRI and IRI groups (Figure 6B,C). Moreover, the administration of IRI at 40 mg/kg over four consecutive days did not decrease the MPO or TNF-α levels in the colon (Figure 6D,E). The colonic microbiota colonization in mice was altered after the administration of both Lipo-IRI and IRI. According to the Illumina 16S metagenomics sequencing data, the intestinal bacterial diversity and richness among the three groups (PBS, IRI, and Lipo-IRI treated groups) showed no difference, but the *Bifidobacterium* spp. content was slightly improved in the Lipo-IRI-treated group (Figure 6F–H).

### 2.8. Generation of Targeting Liposome pHCT74-Lipo-IRI and Evaluation of Their Therapeutic Efficacy in Large Tumor-Bearing Mice

The internalization of pHCT74-Lipo-IRI occurred more readily than non-targeted Lipo-IRI (a 3-fold difference) in HCT 116 cells (Figure 7A). Additionally, the in vitro cytotoxicity was compared between Lipo-IRI and pHCT74-Lipo-IRI using an MTT assay. The IC_50_ value of pHCT74-Lipo-IRI was approximately 2-fold lower than that of Lipo-IRI (Figure 7B), indicating that pHCT74-Lipo-IRI was the more toxic formulation. Since the in vitro characteristics of the targeted liposomes were preferable to the non-targeted liposomes, a combination targeted regimen was evaluated in the HCT 116-derived large tumor model (tumor size: 500 mm^3^). Mice were treated with either PBS, IRI + Dox, Lipo-IRI + Lipo-Dox, or the targeted liposomal drug pair (pHCT74-Lipo-IRI + pHCT74-Lipo-Dox) via intravenous (i.v.) injection (twice weekly × 5). The tumor sizes in all groups except the PBS control decreased over the duration of the drug treatment. By day 35, the tumor growth was inhibited by 74.0%, 93.3% and 96.7% in the IRI + Dox, Lipo-IRI + Lipo-Dox and pHCT74-Lipo-IRI + pHCT74-Lipo-Dox groups, respectively (Figure 7C). Interestingly, the mice treated with a combination therapy showed gradual tumor shrinkage until day 38, and did not lose body weight during the drug administration period (Figure 7D). The mice treated with pHCT74-Lipo-IRI + pHCT74-Lipo-Dox exhibited the smallest tumor volume of all the groups after the treatment was stopped on day 31, and this group also had the longest median survival (Figure 7E).

## 3. Discussion

Chemotherapy is one of the most common treatments for cancer; however, patients generally have poor outcomes with severe side effects due to nonspecific drug delivery to non-target organs. The use of liposomal drug delivery systems has been reported to improve drug efficacy and minimize drug-related systemic toxicity. IRI, a Topo I inhibitor, is a first-line drug for CRC patients and has a molecular structure that exists in equilibrium between the lactone and carboxylate states. The IRI^L^ isoform is highly susceptible to hydrolysis, especially in acidic conditions that commonly exist in the tumor microenvironment. Furthermore, IRI^L^ is quickly converted to the inactive IRI^C^ isoform at a physiological pH. To address these issues of stability and potency, we encapsulated IRI in mild acidic conditions to prevent it from a premature degradation. More than 97% encapsulation efficiency was achieved by incorporating a calcium ionophore in the lipid bilayer of Lipo-IRI. The ionophore provided a channel to maintain the transmembrane pH gradient by shuttling copper ions and protons, thereby enhancing the loading efficiency [29,30]. Furthermore, IRI in the liposomal formulation was more than 97% active IRI^L^ due to the low internal pH of the liposomes.

IRI interacted with copper and stabilized the inner structure of the liposome [29,31]. The Lipo-IRI particle size (105 nm) and loading concentration (64,000 ± 8900 IRI molecules/liposome) were confirmed by a Nanoparticle Tracking Analysis. The IRI salt was precipitated within the liposome, as observed with cryo-TEM. Moreover, the drug release profile revealed that the formulation is stable at a neutral pH and the drug is released in acidic conditions. Lipo-IRI particles prolong the circulation time in the blood and improve the pharmacokinetic and bio-distribution of their encapsulated drugs. After an intravenous administration, the liposome particles are large enough to be excluded from the normal endothelium. In solid tumors, the angiogenic tumor vasculature becomes leakiness that particulate liposomes can extravasate and localize in the tissue interstitial space, making it possible for more drug delivering liposomes to accumulate within the tumor by enhanced permeability and retention (EPR) effect [32,33]. Upon arrival in the tumor tissues, the liposomes are internalized by the tumor cells, fused with the low pH compartments of the endosomes, and subsequently broken down to release encapsulated drugs into the intracellular space of the cells.

The pharmacokinetic profiling indicated that Lipo-IRI shows improved pharmacokinetics and prolongs the circulation time compared to free IRI. While IRI was eliminated rapidly through the mononuclear phagocyte system, Lipo-IRI exhibited a much more desirable half-life and AUC. The clearance for Lipo-IRI was decreased by 4.8-times compared to that of free IRI, suggesting that Lipo-IRI has a longer residence time in the blood stream. IRI was released slowly from the liposome vesicles after Lipo-IRI entered into systemic circulation, and a similar phenomenon has been reported in previous papers [26,34]. The Lipo-IRI accumulation in the tumor was also better than for free IRI. Our in vitro and in vivo findings indicated that tumor cell lines with a high carboxylesterase enzyme activity were sensitive to IRI. Indeed, IRI is a prodrug that can be converted to its active metabolite (SN-38) by carboxylesterase. The abundance of carboxylesterase enzyme in liver and intestine was investigated, and recent reports demonstrated that tumor tissues have a high expression of the carboxylesterase enzyme [35,36,37].

Significant decreases in tumor size were observed in the Lipo-IRI-treated HCT 116 xenograft models. Compared to the PBS-treated mice, which had 1418.22 ± 190.11 nm tumors, the tumor sizes shrank to 466.33 nm ± 183.91 nm and 48.65 ± 39.35 nm in the 2 mg/kg and 5 mg/kg twice weekly Lipo-IRI-treated mice, respectively. These small tumors indicate that the mice were almost cured, and the tumor size continued to decrease for a week after terminating the drug administration, with the tumor growth inhibition reaching 96%. The tumor-bearing mice treated with free IRI had no significant tumor growth inhibition, while Lipo-IRI exhibited improved pharmacokinetics and greatly decreased the tumor mass.

Two major side effects of IRI are diarrhea and neutropenia. [38] IRI-associated diarrhea can be severe and clinically significant, sometimes leading to dehydration that requires the patient to be hospitalized [39,40]. The dose-limiting toxicity of IRI was reported in previous papers, and it has been shown that diarrhea can be improved by altering dosing schedules, inhibiting the enzyme activity and inducing or modifying the intestinal microflora [41,42,43,44]. Imbalances of electrolytes are known to be crucial for causing diarrhea, and PGE2 has been reported to stimulate colonic secretion, interfere with Na^+^, K^+^, -ATPase activity, and reduce absorption capacity [42,45]. In our study, the colons of the mice treated with Lipo-IRI twice weekly had a diminished PGE2 level. MPO is an inflammatory marker that is abundant in neutrophil granulocytes and is only found in inflamed tissue [46]. Our data showed that the MPO and TNF-α levels were not significantly different between drug treatments and control groups, regardless of the dosing regimen. The results implied that a 5 mg /kg dose of Lipo-IRI had potential anti-tumor activity without causing colonic inflammation. The monocyte and lymphocyte content in the Lipo-IRI group was recovered, and no severe hepatotoxicity or nephrotoxicity was observed. It is worth noting that the colon images of the IRI- or Lipo-IRI-treated mice were slightly inflamed, but there was no obvious crypt damage detected by histology. Furthermore, the *Bifidobacterium* spp. level was elevated in the Lipo-IRI-treated group, indicating that beneficial microbiota were present in the intestine.

A limited blood supply and high interstitial fluid pressure have occurred in large tumors [26,47,48], resulting in a poor uptake of anti-cancer drugs. To evaluate whether the targeted liposomes could enhance the therapeutic efficacy, the pHCT74-targeted liposomes were used to treat a mouse with a large tumor. Two types of anti-cancer drugs (IRI: Topo I inhibitor, and Dox: Topo II inhibitor) were utilized as candidate drugs and incorporated into liposomes with a targeting peptide (pHCT74) that is specific for CRC cell lines. Compared with Lipo-IRI, pHCT74-Lipo-IRI enhanced the drug delivery and showed increased the toxicity to HCT 116 cells. The co-administration of pHCT74-Lipo-IRI and pHCT74-Lipo-Dox had the best anti-tumor efficacy and median survival in the large HCT 116 xenograft model.

Currently, FOLFIRI (FOL: Leucovorin Calcium, F: Fluorouracil, IRI: Irinotecan) is an approved chemotherapy combination regimen used in clinical practice as a major treatment for CRC and sometimes for pancreatic cancer [49,50]. Following an infusion of FOLFIRI, patients often experience several unpleasant dose-limiting side effects. As such, this combination of chemotherapeutic drugs may be improved by formulating liposome-encapsulated versions that reduce toxicity and benefit survival, similar to the successful application of liposomal IRI in metastatic pancreatic cancer treatments [51,52]. Our results show that tumor-bearing mice treated with Lipo-IRI and combinations of liposome-encapsulated IRI and Dox have an excellent anti-tumor efficacy and lower toxicity compared to conventional chemotherapeutic formulations. Moreover, our results suggest that targeted liposomes may also contribute to the improvement of tumor inhibition and to the extension of survival. Thus, targeted and non-targeted liposome-encapsulated formulations of combination treatments, such as FOLFIRI, should be further evaluated in preclinical and clinical studies, especially for the treatment of CRC.

## 4. Materials and Methods

### 4.1. Materials and Animals

Irinotecan hydrochloride (IRI/CPT-11) and 7-Ethyl-10-hydroxy-camptothecin (SN-38) were purchased from ScinoPharm Taiwan Ltd. 1,2-Distearoyl-sn-glycero-phosphocholine (DSPC) and PEG-2000-DSPE (1,2-distearoyl-sn-glycero-3-phosphoethanol-amine-N-[methoxy(polyethylene glycerol)-2000]) were provided from Lipoid GmbH (Ludwigshafen, Germany). PEG-DSPE was obtained from NOF Corporation (Tokyo, Japan). Dulbecco’s Modified Eagle’s Medium (DMEM), RPMI-1640 media, fetal bovine serum (FBS), and penicillin-streptomycin were purchased from Invitrogen (Carlsbad, CA, USA). Methanol, dimetheyl sulfoxide (DMSO), triethyl ammonium (TEA) were obtained from J.T. Baker (Phillipsburg, NJ, USA). The Prostaglandin E2 Parameter Assay Kit, Mouse Myeloperoxidase DuoSet ELISA and Mouse TNF-α DuoSet ELISA Development kit were purchased from R&D systems (Minneapolis, MN, USA). All other chemicals and reagents were purchased from Sigma-Aldrich (St. Louis, MO, USA).

All cells were purchased from ATCC and were cultured according to the protocols from ATCC. HCT 116 (human colorectal carcinoma) and SK-HEP-1 (human hepatic adenocarcinoma) were cultured in DMEM. The A549 (human lung adenocarcinoma) cell line was cultured in RPMI-1640 media. Both types of media were supplemented with 10% FBS, 2 mM L-glutamine, and 1% penicillin-streptomycin (P/S). All animals were maintained in the institutional animal facility with a temperature and humidity control under a 12 h light-dark cycle. The animals were handled in accordance with the guidelines set forth by the Academia Sinica Institutional Animal Care and Utilization Committee (IACUC). The ethical code: 17-12-1179; permission date: start date (1 July 2018), end date (30 June 2021).

### 4.2. Liposomal Drug Preparation

Liposomal doxorubicin (Lipo-Dox) was prepared according to our previously published method [26]. Lipo-IRI was prepared by a standard thin-film hydration method [53]. Briefly, DSPC, cholesterol and PEG-2000-DSPE were dissolved in chloroform at the required molar ratio (DSPC: cholesterol: PEG-2000-DSPE, 3: 2: 0.15) and the solvent was removed by rotary evaporation. The lipid films were hydrated at 60 °C by the addition of a 300 mM copper sulfate solution prior to five freeze-thaw cycles. The multi-lamellar vesicle suspension was extruded through three stacked polycarbonate filters eight times (0.2, 0.1 and 0.08 μm) at 60 °C. The external buffer was replaced by passing the solution through a column packed with Sephadex G-50 Fine (GE Healthcare, Uppsala, Sweden), which had been equilibrated with an SHE buffer (5% sucrose, 5 mM HEPES, 15 mM EDTA), at pH 6.5.

The ionophore A23187 (1 mg/mL solution in 100% ethanol) was pre-incubated with liposomes for 20 min prior to drug loading [29]. Irinotecan hydrochloride was added to the liposome suspension solution at 60 °C according to the required molar ratio (0.8 mol IRI/mol phospholipid). The pHCT74-PEG-DSPE conjugate was synthesized in a stoichiometric molar ratio of 1.1:1 for 72 h, purified with dialysis and lyophilized. Afterward, the pHCT74-PEG-DSPE conjugate was integrated with preformed liposomes for another 30 min. Lipo-IRI and pHCT74-Lipo-IRI were separated by a Sephadex G-50 Fine column equilibrated with HEPES buffer, at pH 6.5 (5 mM HEPES, 145 mM NaCl).

### 4.3. Phosphorus Content Analysis

Phospholipid was quantified by Bartlett’s method [54]. Briefly, Lipo-IRI was dissolved in 10 N H_2_SO_4_ and heated at 170 °C for 30 min. H_2_O_2_ (9% final concentration) was added, and the solution was reheated for at least 30 min. (NH_4_)_6_Mo_7_O_24_·_4_H_2_O (0.22%) was added, followed by the addition of 15% ascorbic acid. The solution was mixed thoroughly prior to heating in boiling water for 10 min. The optical density at 830 nm was recorded by a spectrophotometer (SpectraMax M5, Molecular Device, San Jose, CA, USA).

### 4.4. Characterization of Lipo-IRI

The particle size and zeta potential were determined by Dynamic Light Scattering (DLS, Zetasizer Nano ZS, Malvern Instruments, UK). The Lipo-IRI solution was diluted and transferred to a folded capillary zeta cell (Malvern, UK). The samples were equilibrated for 120 s at 25 °C prior to the size and zeta potential measurements. The hydrodynamic diameter (*z*-average) and zeta potential of Lipo-IRI were analyzed by Zetasizer software, version 7.11 (www.malvern.com). The particle concentration was evaluated by Nanoparticle Tracking Analysis (NTA) using a Nanosight NS300 (Malvern Instruments, Worcestershire, UK) with an auto-sampler. Each sample was passed through a 0.22 μm filter and diluted with buffer until the particle concentration was suitable for measurement. The particle size distribution and concentration were evaluated with NTA 3.0 software (Malvern Instruments, Worcestershire, UK).

The structure of Lipo-IRI particles was analyzed by cryogenic transmission electron microscopy (cryo-TEM, Tecnai F20, Philips, Eindhoven, the Netherlands) [26]. Briefly, Lipo-IRI solution was transferred onto 300-mesh copper grids covered with porous carbon film (HC300-Cu, PELCO) for blotting and plunging in a 100% humidity temperature-controlled chamber by Vitroblot (FEI). The copper grids were stored under liquid nitrogen (LN_2_) and transferred to the electron microscope on a cryo-stage for imaging.

### 4.5. Quantification of Lipo-IRI

The IRI concentration was measured by RP-HPLC (Wates Instruments, Milford, MA, USA) [55]. Each sample was dissolved in acidified isopropanol (80 mM HCl in isopropanol) and diluted to an appropriate concentration prior to injection into a C18 column (Atlantis T3; 5.0 μm; 4.6 mm × 100 mm column, Waters). The fluorescence detection (Waters 2475 multi λ fluorescence detector) was carried out at an excitation wavelength of 375 nm and an emission wavelength of 500 nm. The products were eluted by a gradient mobile phase of acetonitrile (20–80%) in buffer (3% triethylammonium acetate, pH 5.5), at a flow rate of 0.9 mL/min; the total running time was 15 min. The system control and RP-HPLC chromatography data analysis were performed using Empower 2.0 software. The IRI encapsulation efficiency was calculated with the following formula:(1)$$\mathrm{Encapsulation}\;\mathrm{efficiency}\;(\%)=\{[\frac{IRI}{PL}{]}_{p}/[\frac{IRI}{PL}{]}_{i}\}\times 100\%$$
where [$\frac{IRI}{PL}$]_p_ refers to the IRI content drug-to-lipid ratio following isolation of the encapsulated liposomes, and [$\frac{IRI}{PL}$]_i_ refers to the IRI content drug-to-lipid ratio of the initial preparation prior to IRI loading.

### 4.6. In Vitro Release Profile

An aliquot of Lipo-IRI was transferred into a dialysis bag with a molecular weight cut-off of 10 kDa (Slide-A-Lyzer MINI Dialysis unit, Thermo scientific, Rockford, IL, USA), and incubated in HEPES at distinct pH conditions (pH 4.0, pH 5.0 and pH 6.5). The dialysate was placed on a magnetic stirrer, rotating at 150 rpm, and the temperature was maintained at 37 °C with protection from light throughout the experiment. The samples were withdrawn periodically, and the amount of drug released into the buffer was quantified by an RP-HPLC system at various times up to 72 h. The samples were collected rapidly and stored at −20 °C until the RP-HPLC analysis. The amount of IRI released into the buffer was analyzed according to following equation:(2)$$\mathrm{The}\;\mathrm{percentage}\;\mathrm{of}\;\mathrm{drug}\;\mathrm{released}\;(\%)=\frac{M_{t}}{M_{0}}\times 100\%$$
where *M_t_* is the amount of drug released at time *t*, and *M*_0_ is the total amount of drug in the liposome.

### 4.7. Cell Viability Assay

The tumor cell lines were seeded (5000 cells/well) in 96-well plates (Corning Inc., Corning, NY, USA) in DMEM or RPMI-1640 media (with 10% FBS, 1% P/S, and 2 mM L-glutamine). After 24 h incubation at 37 °C, the IRI treatments were administered at concentrations ranging from 0.19 to 100 μM. The treated cells were incubated for another 48 h at 37 °C. Next, 100 μL of 0.1 mg/mL MTT [3-(4,5-dimethylthiazol-2-yl)-2,5-diphenyltetrazolium bromide] reagent was added to each well and incubated for 2 h. The media was removed from each well prior to the addition of 200 μL DMSO to dissolve the formazan crystals. The absorbance at 570 nm was read on a spectrophotometer and the half maximal inhibitory concentration (IC_50_) was determined by a data analysis with GraphPad Prism 6.0 software (San Diego, CA, USA).

### 4.8. Pharmacokinetics and Bio-Distribution Study

IRI or Lipo-IRI were administered to tumor-bearing mice via tail vein injections at doses of 30 mg IRI/kg. The blood samples were collected at various times post-injection via orbital sinus sampling (*t* = 0.083, 0.16, 0.33, 0.66, 1, 3, 7, 9, 11, 24, 48 and 72 h after IRI or Lipo-IRI administration). The plasma was obtained by the centrifugation of the blood at 3000× *g* for 10 min. The samples were deproteinized by adding cold acidic isopropanol prior to the centrifugation of the mixture at 10,000 × *g* for 10 min. The upper layer solution was collected and stored at −20 °C before quantitation. The organs (brain, heart, lung, liver, kidney, spleen, colon and tumor) were harvested after the administration of IRI or Lipo-IRI for 0.083, 0.16, 1, 4, or 24 h, and were perfused with a PBS buffer. All tissues were homogenized and stored at −80 °C prior to the analysis. The concentrations of IRI and Lipo-IRI were determined by RP-HPLC, and the pharmacokinetic parameters were calculated with GraphPad Prism 6.0 software.

### 4.9. In Vivo Anti-Tumor Efficacy in Small Tumor Models

Non-obese diabetic/severe combined immunodeficiency (NOD/SCID) mice (4–6 weeks old) were injected subcutaneously (s.c.) in the dorsolateral flank with 10^6^ tumor cells in a serum-free medium. Mice with size-matched tumors (approximately 100 mm^3^) were randomly assigned to different treatment groups and injected intravenously with an IRI, Lipo-IRI, or PBS administration via the tail vein. The drugs were administrated by intravenous injection twice weekly over 4 weeks (twice weekly × 4) (2, 5 or 10 mg/kg for HCT 116 model; 2 or 5 mg/kg for SW620 model). The mouse body weight, tumor mass and survival rate were measured. The tumor volumes were calculated using the following formula: tumor volume = length × (width)^2^ × 0.52. The data are presented as the mean ± standard error of the mean. The tumor growth inhibition (TGI) was determined by the formula:(3)$$\mathrm{TGI}=[1-(\frac{V_{Treated}}{V_{Vehicle}})]\times 100\%,$$
where *V_Vehicle_* and *V_Treated_* are defined as the mean tumor volume (mm^3^) of the vehicle group and test group at time *t*, respectively.

### 4.10. Toxicity and Side Effects

The tumor-bearing mice were euthanized at the end of treatment; the blood and serum were collected for complete blood count, hepatotoxicity and nephrotoxicity analyses. The colon was washed with phosphate buffered saline (PBS) and homogenized (MICCRA) prior to centrifugation. The total protein concentration was assessed by the bicinchoninic acid (BCA) assay. The amounts of PGE2, MPO and TNF-α in the tissue extracts were determined by commercial ELISA assays, according to the manufacturer’s instructions. The levels of PGE2, MPO and TNF-α were normalized to the protein concentration of each sample and the results are reported as microgram per gram of protein (μg/g). Additionally, the MPO and TNF-α productions in the colon were determined after the drug treatment. The colon and fecal material were harvested for a cytokine analysis and microbial community analysis, respectively. The fecal samples were collected and snap-frozen in LN_2_ and allocated for 16S metagenomics sequencing (Illumina MiSeq system, San Diego, CA, USA).

The animals were euthanized after drug treatment for a histological study. The colon samples were harvested, washed with PBS and fixed with 4% neutral buffered paraformaldehyde, after which the tissues were snap frozen in an optimal cutting temperature (OCT) compound. The blocks were sectioned at a thickness of 10 μm and stored at −80 °C prior to the histological analysis. The other organs (brain, heart, liver, kidney and spleen) were fixed and embedded in paraffin. The tissue sections were stained with hematoxylin and eosin (H&E) by standard protocols.

### 4.11. Combination Targeted Therapy in Large Tumor Model In Vivo

The tumor cells were injected s.c. into the dorsolateral flank of NOD-SCID mice (4–6 weeks old). The tumor bearing mice were randomly divided into four treatment groups when the tumor sizes reached 500 mm^3^. The treatment groups were: (1) PBS, (2) free drug (IRI+Dox), (3) liposome-encapsulated drug (Lipo-IR+Lipo-Dox) and (4) targeted liposomal drug (pHCT74-Lipo-IRI+pHCT74-Lipo-Dox). The drugs were administered via tail vein injection as single drugs or combinations (5 mg IRI/kg and 1 mg Dox/kg) twice weekly. The tumor volume, body weight and survival rate were monitored. The tumor growth inhibition was estimated by the above formula.

### 4.12. Statistical Analysis

The data are presented as the mean ± SD (standard deviation) or SEM (standard error of the mean) of at least 3 independent experiments, unless otherwise noted. An unpaired Student’s *t*-test was used for a comparison of the two independent groups and was performed using GraphPad Prism 6.0 software. To assess differences in survival between the groups, a log-rank analysis was used. *p*-values less than 0.05 were considered statistically significant.

## 5. Conclusions

In summary, the results of this study indicate that Lipo-IRI can be produced with a high loading efficiency and stability during long-term storage. Moreover, Lipo-IRI is a potent toxin to colon, liver and lung tumor cell lines, with improved pharmacokinetic parameters and enhanced in vivo therapeutic efficacy compared to free IRI. The entrapment of IRI in liposomes may offer a safer and more effective therapeutic strategy than traditional formulations. Further improvements in therapeutic efficacy and patient outcomes may be possible by applying a drug regimen with a combination of targeted liposome-encapsulated chemotherapeutics. Additionally, the Lipo-IRI treatment did not disrupt the population of beneficial *Bifidobacterium* spp. compared to a IRI treatment, possibly indicating that the new formulation will cause fewer side effects. This new formulation of Lipo-IRI increases the therapeutic efficacy of cancer treatment by increasing drug accumulation in the tumor tissue, which reduces side effects by decreasing damage to normal tissues.

## Figures and Tables

**Figure 1 cancers-11-00281-f001:**
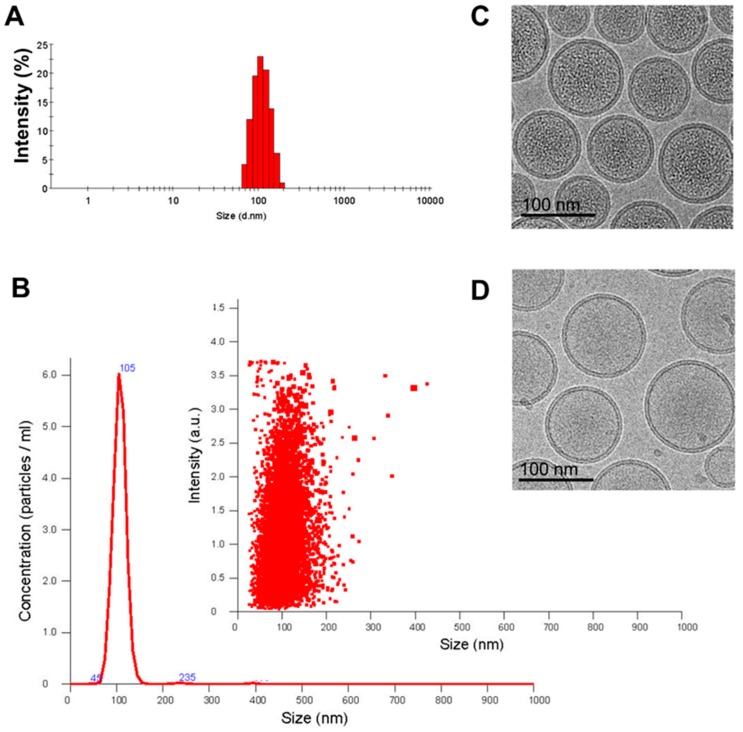
Characterization of Lipo-IRI (irinotecan) formulation. The size of Lipo-IRI was estimated using (**A**) dynamic light scattering (DLS) and (**B**) nanoparticle tracking analysis (NTA). The size distribution and particle concentration were calculated as intensity and particles/ml, respectively. The cryo-transmission electron microscopy (TEM) imaging was performed on a drug-loaded 1,2-Distearoyl-sn-glycero-phosphocholine (DSPC)/Cholesterol formulation either (**C**) with or (**D**) without irinotecan (IRI). (10 μmol/mL phospholipid; scale bar represents 100 nm).

**Figure 2 cancers-11-00281-f002:**
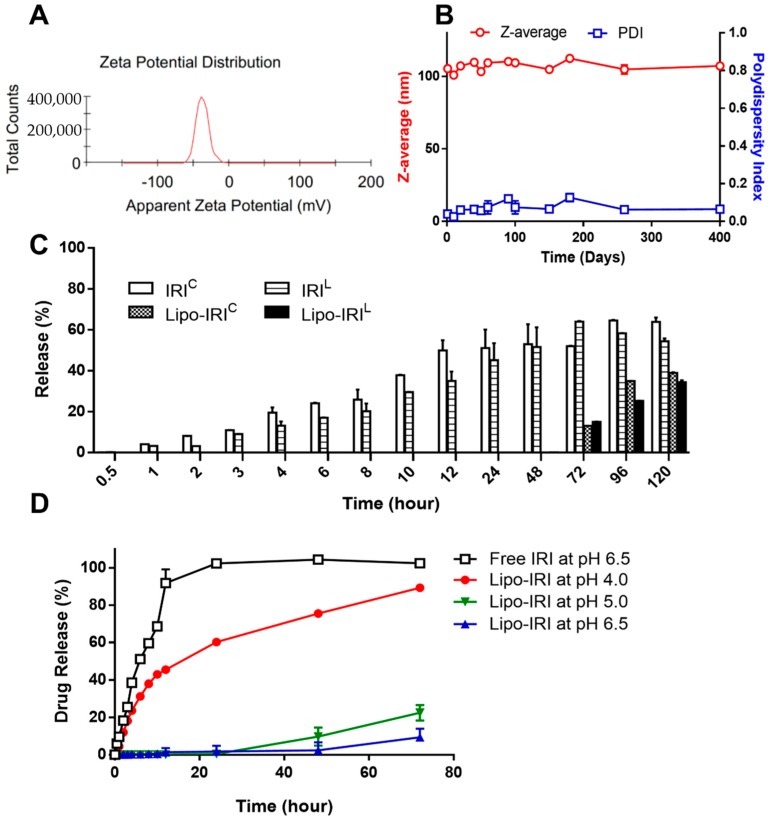
Evaluation of stability of Lipo-IRI. (**A**) The zeta potential was assessed by DLS. (**B**) The z-average and polydispersity index (PDI) were estimated after 400 days of storage using a DLS instrument. (**C**) Free IRI and Lipo-IRI were dialyzed in a HEPES buffer at pH 7.4. The contents of IRI^L^ and IRI^C^ in the dialysate were quantified by reversed phase-high performance liquid chromatography (RP-HPLC). (**D**) The IRI and the Lipo-IRI suspension were also dialyzed in a HEPES buffer at pH 4.0, 5.0 and 6.5. The amount of IRI in the dialysate was analyzed. The data are presented as the mean ± SD (*n* = 3).

**Figure 3 cancers-11-00281-f003:**
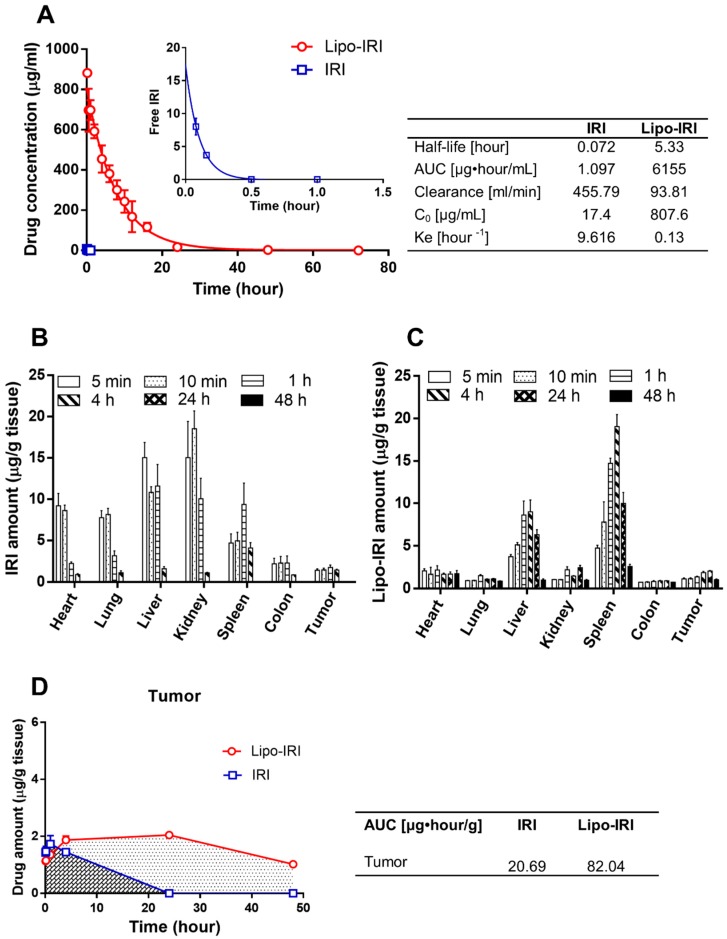
Pharmacokinetic and bio-distribution profile of Lipo-IRI and free IRI in tumor-bearing mice. Following an injection with IRI or Lipo-IRI, the blood and organs were collected at various time-points. The pharmacokinetics are shown in (**A**). The inset shows the profiles in the first hour after administration. The data are shown for Lipo-IRI (○) and free IRI (□). The organs were homogenized, and the amount of IRI was quantified by RP-HPLC. The bio-distribution profiles of IRI and Lipo-IRI are shown in (**B**) and (**C**). The AUC for Lipo-IRI and IRI is shown in (**D**). The data are represented as the mean ± SD (*n* = 3).

**Figure 4 cancers-11-00281-f004:**
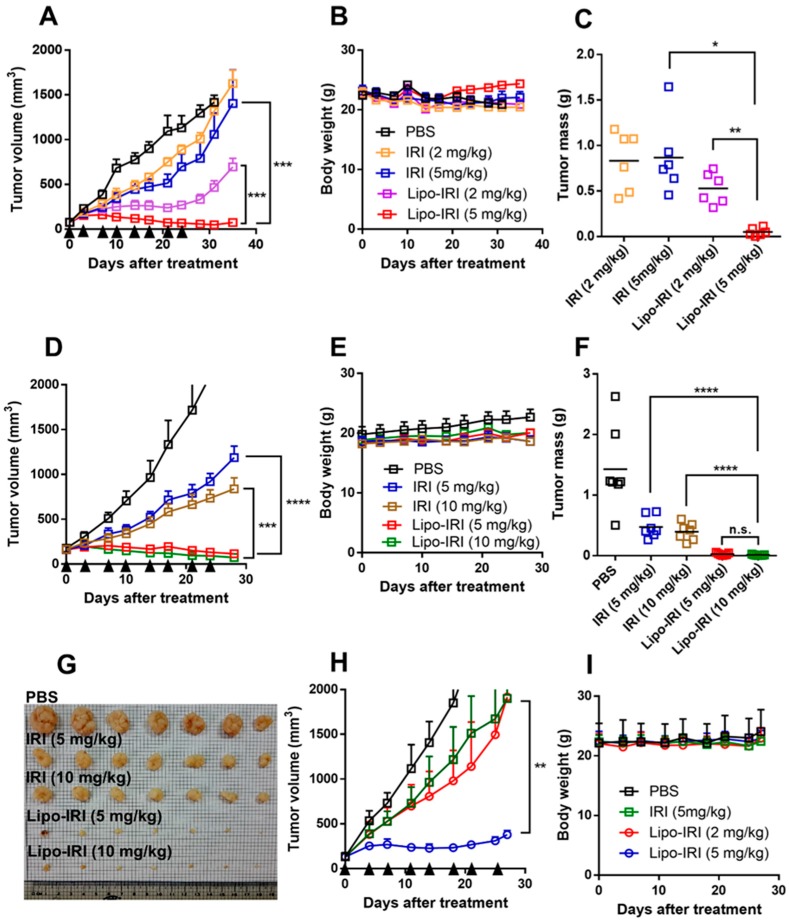
Lipo-IRI effectively reduced tumor volume and mass in colorectal cancer (CRC) xenograft models. HCT 116 (**A**–**G**) or SW620 (**H**,**I**) tumor-bearing mice received i.v. doses (arrows) of PBS, IRI or Lipo-IRI (2 or 5 mg/kg) twice per week. In the HCT 116 model, (**A**) the tumor volume and (**B**) body weight were monitored during the treatment. (**C**) The tumor mass was measured after sacrificing the mice. The data are presented as the mean ± SEM (*n* = 6). In another experiment, tumor bearing mice were treated with PBS, IRI or Lipo-IRI (5 or 10 mg/kg) twice weekly. (**D**) The tumor volume and (**E**) body weight were monitored. (**F**,**G**) The tumor mass was measured after sacrificing the mice (*n* = 7). In the SW620 xenograft model, mice were administered IRI or Lipo-IRI twice weekly, and the tumor volume and body weight were recorded (**H** and **I**). The data are presented as the mean ± SEM. * *p* < 0.05, ** *p* < 0.01, *** *p* < 0.001 and **** *p* < 0.0001.

**Figure 5 cancers-11-00281-f005:**
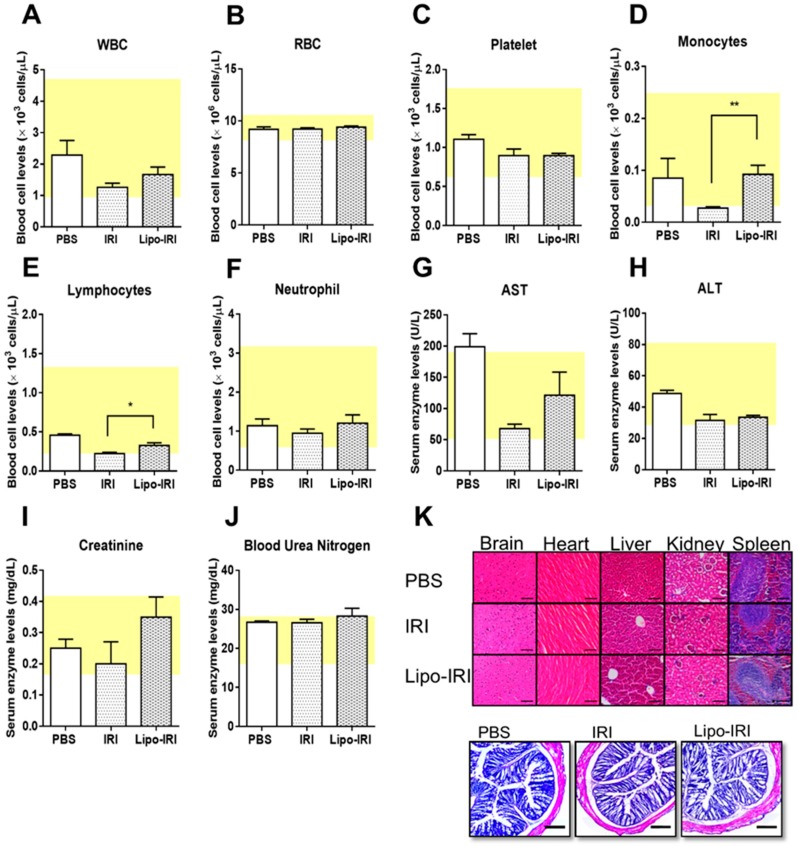
Evaluation of side effects in hematological and histological study. The mice were treated with either IRI or Lipo-IRI at 5 mg/kg twice weekly. The plasma and serum were harvested and analyzed. The hematology for (**A**) white blood cell (WBC), (**B**) red blood cell (RBC), (**C**) platelet, (**D**) monocyte, (**E**) lymphocyte, and (**F**) neutrophil are shown. The (**G**) aspartate aminotransferase (AST), (**H**) alanine aminotransferase (ALT), (**I**) creatinine, and (**J**) blood urea nitrogen levels were measured to assess the hepatotoxicity and nephrotoxicity. The data are presented as the mean ± SEM (*n* = 3). The yellow box indicates the normal range. Compared to IRI, * *p* < 0.05. (**K**) The major organs were harvested for paraffin embedding, and the colon was frozen in optimal cutting temperature (OCT) compound after the end of the treatment. Tissue sections were stained with H&E for a histological analysis (Scale bars, 200 μm).

**Figure 6 cancers-11-00281-f006:**
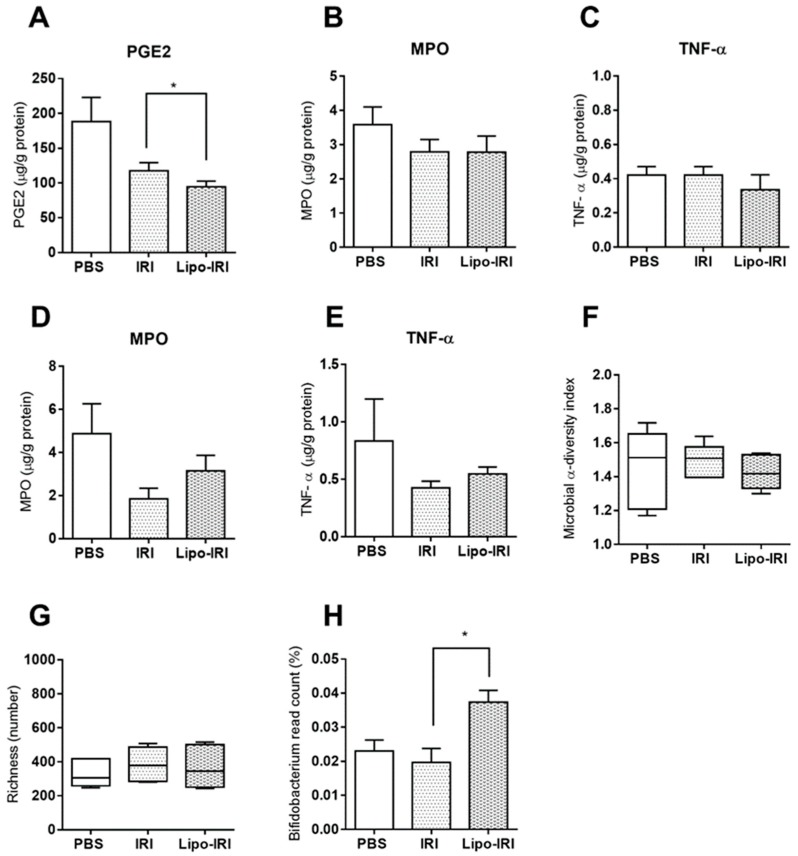
Quantification of inflammatory mediators in colon tissues and microbiota analysis in fecal samples. The colon tissues were harvested after the drug treatment and homogenates were assayed with sandwich ELISA for PGE2, MPO and TNF-α (**A**, **B** and **C**, respectively). The MPO and TNF-α levels in the colon were assessed after four consecutive treatments (**D** and **E**). The microbiota diversity, richness, and *Bifidobacterium* spp. population are shown in (**F**), (**G**) and (**H**). Each column represents the mean ± SD (*n* = 3). Compared to IRI, * *p* < 0.05.

**Figure 7 cancers-11-00281-f007:**
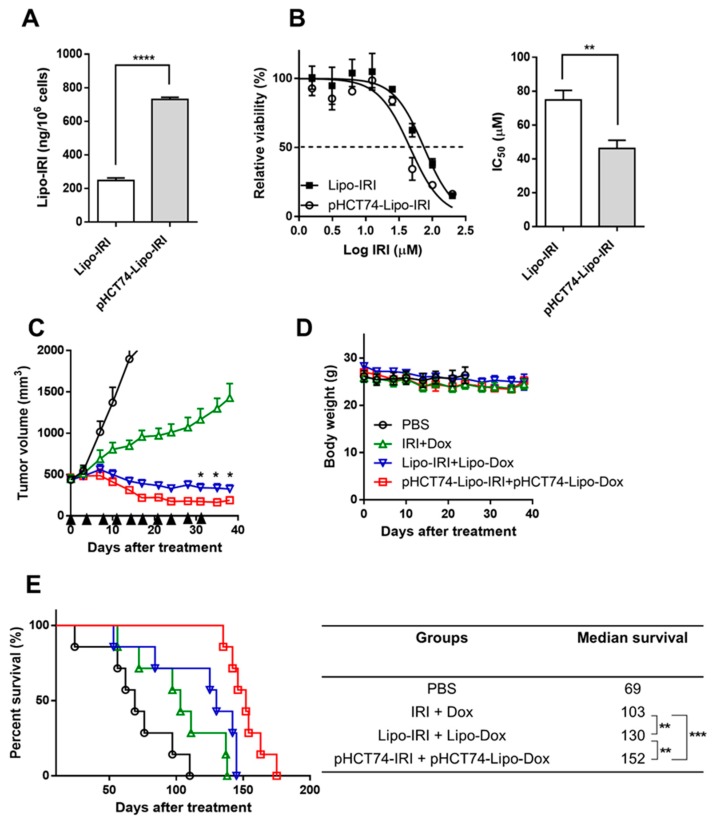
Enhanced effect of targeted liposome-encapsulated treatment combination in an hCRC large tumor model. The drug internalization (**A**) and cytotoxicity (**B**) of pHCT74-Lipo-IRI were compared with non-targeted Lipo-IRI in vitro. The tumor bearing mice (500 mm^3^) received PBS, IRI + Dox, Lipo-Dox + Lipo-IRI, pHCT74-Lipo-IRI + pHCT74-Lipo-Dox 10 times over the course of 5 weeks (twice weekly × 5). The tumor volume and body weight are shown (**C** and **D**). A Kaplan-Meier curve and median survival of each group are shown (**E**). The data were analyzed by GraphPad Prism 6.0 software (*n* = 7). * *p* < 0.05, ** *p* < 0.01, *** *p* < 0.001 and **** *p* < 0.0001.

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
