# Peer review of "Liposomal Irinotecan for Treatment of Colorectal Cancer in a Preclinical Model"

_cancers, 2019, doi:10.3390/cancers11030281_

Reviewer 1 Report

In the manuscript titled, “Liposomal irinotecan for treatment of colorectal cancer in a preclinical model”, Huang et al focused on the potential of liposomal irinotecan for the treatment of colorectal cancer. The authors demonstrated that formulation of liposomal irinotecan has longer half-life and better outcome compared to irinotecan in solution utilizing in-vitro and in-vivo experiments and indicated that liposomal irinotecan has potential as a therapeutic agent for the treatment of colorectal cancer. The manuscript is well-written and provided control experiments to confirm their results. The manuscript would be interesting to the target audience and would be an important addition to the journal. The authors are recommended to check the description of some of the figure legends and to add an overall conclusion in the discussion section of the manuscript.

Good luck

Author Response

Response:

We thank the reviewer for these suggestions. The Figure Legends have been revised as suggested:Figure 3, Page 6; Figure 4, page 7; Figure 5, page 8; Figure 6, page 9; Figure 7, page 11.

We have added the following information to the Discussion of the revised manuscript (Discussion section, page 13) as suggested:

“Currently, FOLFIRI (FOL: Leucovorin Calcium, F: Fluorouracil, IRI: Irinotecan) is an approved chemotherapy combination regimen used in clinical practice as a major treatment for CRC and sometimes for pancreatic cancer [49,50]. Following infusion of FOLFIRI, patients often experience several unpleasant dose-limiting side effects. As such, this combination of chemotherapeutic drugs may be improved by formulating liposome-encapsulated versions that reduce toxicity and benefit survival, similar to the successful application of liposomal IRI in metastatic pancreatic cancer treatment [51,52]. Our results show that tumor-bearing mice treated with Lipo-IRI and combiantions of liposome-encapsulated IRI and Dox have excellent anti-tumor efficacy and lower toxicity compared to conventional chemotherapeutic formulations. Moreover, our results suggest that targeted liposomes may also contribute to improvement of tumor inhibition and extension of survival. Thus, targeted and non-targeted liposome-encapsulated formulations of combination treatments, such as FOLFIRI, should be further evaluated in preclinical and clinical studies, especially for treatment of CRC.”

Reviewer 2 Report

The authors are to be congratulated on their valuable study.

In clinical practice, FORFIRI (IRI in combination with 5-FU/LV) is the established treatment for colorectal cancer. Please address the issue of future phase 2-3 clinical trials comparing free IRI with Lipo-IRI in combination with 5-FU/LV, as the basic studies led to clinical trials followed by approval of Lipo-IRI in metastatic pancreatic cancer (Wang-Gillam A et al. European Journal of Cancer 2019; 108: 78-87).

Author Response

We have added several sentences to the Discussion of the revised manuscript (Discussion section, page 13) as suggested:

“Currently, FOLFIRI (FOL: Leucovorin Calcium, F: Fluorouracil, IRI: Irinotecan) is an approved chemotherapy combination regimen used in clinical practice as a major treatment for CRC and sometimes for pancreatic cancer [49,50]. Following infusion of FOLFIRI, patients often experience several unpleasant dose-limiting side effects. As such, this combination of chemotherapeutic drugs may be improved by formulating liposome-encapsulated versions that reduce toxicity and benefit survival, similar to the successful application of liposomal IRI in metastatic pancreatic cancer treatment [51,52]. Our results show that tumor-bearing mice treated with Lipo-IRI and combinations of liposome-encapsulated IRI and Dox have excellent anti-tumor efficacy and lower toxicity compared to conventional chemotherapeutic formulations. Moreover, our results suggest that targeted liposomes may also contribute to improvement of tumor inhibition and extension of survival. Thus, targeted and non-targeted liposome-encapsulated formulations of combination treatments, such as FOLFIRI, should be further evaluated in preclinical and clinical studies, especially for treatment of CRC.”

 Finally, we would like to express our sincere appreciation to the reviewers and the editors for their valuable time in the careful review of our work and for their insightful suggestions. Their comments and suggestions have helped to greatly improve the quality of our work.